# The In Situ Preparation of Ni–Zn Ferrite Intercalated Expanded Graphite via Thermal Treatment for Improved Radar Attenuation Property

**DOI:** 10.3390/molecules28104128

**Published:** 2023-05-16

**Authors:** Ning Xiang, Zunning Zhou, Xiaoxia Ma, Huichao Zhang, Xiangyuan Xu, Yongpeng Chen, Zerong Guo

**Affiliations:** State Key Laboratory of Explosion Science and Technology, Beijing Institute of Technology, Beijing 100081, China; 3120200295@bit.edu.cn (N.X.); zzn@bit.edu.cn (Z.Z.); zhc_1514@126.com (H.Z.); 3220200201@bit.edu.cn (X.X.); 3120195113@bit.edu.cn (Y.C.)

**Keywords:** expanded graphite, Ni–Zn ferrite, intercalation structure, radar wave attenuation

## Abstract

The composites of expanded graphite (EG) and magnetic particles have good electromagnetic wave attenuation properties in the centimeter band, which is valuable in the field of radar wave interference. In this paper, a novel preparation method of Ni–Zn ferrite intercalated EG (NZF/EG) is provided in order to promote the insertion of Ni–Zn ferrite particles (NZF) into the interlayers of EG. The NZF/EG composite is in situ prepared via thermal treatment of Ni–Zn ferrite precursor intercalated graphite (NZFP/GICs) at 900 °C, where NZFP/GICs is obtained through chemical coprecipitation. The morphology and phase characterization demonstrate the successful cation intercalation and NZF generation in the interlayers of EG. Furthermore, the molecular dynamics simulation shows that the magnetic particles in the EG layers tend to disperse on the EG layers rather than aggregate into larger clusters under the synergy of van der Waals forces, repulsive force, and dragging force. The radar wave attenuation mechanism and performance of NZF/EG with different NZF ratios are analyzed and discussed in the range of 2–18 GHz. The NZF/EG with the NZF ratio at 0.5 shows the best radar wave attenuation ability due to the fact that the dielectric property of the graphite layers is well retained while the area of the heterogeneous interface is increased. Therefore, the as-prepared NZF/EG composites have potential application value in attenuating radar centimeter waves.

## 1. Introduction

With the development of radar technology, guided weapons pose a significant threat to critical strategic targets, such as aircraft, missiles, and ships [1,2]. Therefore, protecting one’s own equipment and jamming enemy radars are essential tasks in modern battlefields. As a radar jamming technique, passive interference materials are widely used due to their flexibility, convenience, and high cost-effectiveness ratio.

Passive interference materials commonly include expanded graphite (EG) [3,4,5], graphene [6], carbon nanotubes [7], carbon fibers [8], and conductive polymers [9]. EG, in particular, is a worm-like material formed by graphite intercalation compounds (GICs) through thermal treatment, exhibiting corrosion resistance and good stability. Its excellent dielectricity and rich porous structure make it an ideal material for millimeter-wave radar attenuation through electrical loss [10,11,12,13]. Moreover, EG can be dispersed in the air to form aerosols due to its low density, resulting in target obscuration and large radar attenuation areas. However, for centimeter-wave radar (such as 2–18 GHz) used for search, track, and fire control, EG’s interference effectiveness is insufficient. Magnetic materials can compensate for this deficiency due to their high magnetic loss property [14].

At present, several studies have been conducted on magnetic particles and EG composites that exhibit excellent electromagnetic wave attenuation performance in the centimeter band [15,16]. However, most previous studies involved the direct mixing of EG with ferrite precursors [17] or metal salts [18], which led to poor uniformity of magnetic particles in EG since most of these particles were distributed on the surface of EG. Moreover, the magnetic properties of ferrites are significantly weakened when the ambient temperature is higher than the Curie temperature of ferrites (e.g., 280–450 °C) [19], which can adversely affect the electromagnetic attenuation performance of the composites. In addition, these research results are mainly applied to absorbing coatings rather than aerosol attenuation.

Therefore, to promote the insertion of magnetic particles into the interlayers of EG and ensure that the composites maintain excellent magnetic properties at high temperatures, this study involves embedding ferrites in EG layers, which offers several benefits. Firstly, EG, being the carrier of ferrites, can increase the dispersion area and hang time of aerosols. Secondly, the uniform distribution of ferrites in EG layers can improve the electromagnetic attenuation performance of composites. More importantly, ferrite precursors intercalated with graphite can be combined with pyrotechnics in applications, which then produce ferrites intercalated with EG with the help of the instantaneous high temperature generated by the pyrotechnic reagent explosion, ensuring that the composites maintain good magnetism when used.

In this work, the ferrite precursors are embedded in graphite layers through chemical coprecipitation and the NZF magnetic particles between the EG layers are in situ prepared through subsequent thermal treatment. The morphology and composition of the precursors and NZF/EG composite are analyzed in detail, while the distribution tendency of NZF magnetic particles between graphite layers is calculated by molecular dynamics simulation. The attenuation mechanism of electromagnetic waves is revealed and the main factor affecting the attenuation property of the NZF/EG composite is obtained.

## 2. Results and Discussion

### 2.1. In Situ Formation of NZF Intercalated EG

The preparation diagram of NZF/EG is illustrated in Figure 1. Initially, NG is treated with acids and oxidants to produce GO. GO is mixed with metallic nitrates, resulting in ${\mathrm{Ni}}^{2+}$, ${\mathrm{Zn}}^{2+}$, and ${\mathrm{Fe}}^{3+}$ entering the graphite layers and forming coordination bonds with carboxyl groups on the graphite layers [20]. These metal ions are then transformed into metal hydroxides by ammonia which serve as the precursors of NZF. During the thermal treatment, graphite expands while the metal hydroxides produce NZF, culminating in the formation of NZF/EG. To demonstrate the in situ formation of NZF intercalated EG, various characterizations were employed.

The SEM images and EDS spectra of NG and NZFP/GICs are presented in Figure 2. The results indicate that the surface and edge areas of NZFP/GICs are rougher compared to NG and that the distance between the layers is slightly increased. It is noteworthy that the graphite still exhibits a multilayer structure. Furthermore, particles are present on the surface and edge of NZFP/GICs. EDS spectra analysis demonstrates that these particles contain Ni, Zn, Fe, and Mn elements. The presence of Mn can be explained by the small Mn residue because potassium permanganate is used as an oxidant. The stronger Ni, Zn, and Fe peaks at the edge of graphite reflect the higher content of these metal elements than that of the surface, which indicates that the edge is the primary channel for metal ions to enter the graphite layers.

The XRD patterns of NG and NZFP/GICs are presented in Figure 3a. It is evident that the characteristic peaks of NG on the (002) and (004) crystal planes are located at 26.6°and 54.7°, respectively, suggesting that NG possesses an orderly structure and high crystallinity. Following the intercalation reaction, some of the graphite maintains the original characteristic peaks, e.g., peak 2 and peak 4, while the (002) crystal plane peaks split into two peaks, marked as peak 1 (25.2°) and peak 3 (29.8°) in the curve of NZFP/GICs. The XRD characteristic peaks are indicative of the graphite layer spacing. By utilizing Bragg’s equation [21], the layer spacing ($d_{H}$) of NG and NZFP/GICs is obtained as follows:(1)$$2d_{H}\sin\;\theta =n\lambda$$
where *θ* is the Bragg angle and 2*θ* is the diffraction angle. *n* is the diffraction order and *λ* is the X-ray wavelength. In this specific case, *n* is equal to 1 and *λ* is equal to 0.154056 nm.

The $d_{H}$ of NG and NZFP/GICs are 0.335 nm and 0.353 nm, respectively, indicating a slight increase in the layer spacing of graphite, confirming the results of Figure 2b. Metal ions commonly undergo strong hydrolysis in aqueous solutions and exist in a hydroxy-hydration state. Previous research has shown that the hydrating ionic radii of ${\mathrm{Ni}}^{2+}$, ${\mathrm{Zn}}^{2+},$ and ${\mathrm{Fe}}^{3+}$ are 0.072 nm, 0.066 nm, and 0.074 nm, respectively [22,23,24]. Therefore, the sizes of metal ions in aqueous solution are smaller than the layer spacing of NG and NZFP/GICs, suggesting that NG and NZFP/GICs provide suitable conditions for metal ions to enter the graphite layers, thus confirming the rationality of the intercalation reaction. However, NG’s poor hydrophilicity makes it impractical for the intercalation reaction, so acid and oxidant treatments are performed to improve this property.

The FTIR spectra of NG and NZFP/GICs in the range of 4000–500 cm^−1^ are shown in Figure 3b. The stretching vibration and the bending vibration of hydroxyl groups in adsorbed water are represented by 3439.5 cm^−1^ and 1628.7 cm^−1^, respectively [25]. The absorption peak at 1117 cm^−1^ indicates the stretching vibration of C-O-C, implying defects in the six-membered ring of graphite. The intensity of the infrared absorption peak at 1384 cm^−1^ in NZFP/GICs is notably higher than that of NG, indicating the presence of organic carbonates in graphite [26]. These oxygen-containing groups indicate that graphite has been successfully oxidized. As is well-known, hydrogen bonds form between hydrogen and oxygen elements in these groups, playing a supportive role between graphite layers and leading to an increase in graphite layer spacing [27]. Moreover, coordination bonds are formed between oxygen-containing groups and hydroxy-hydrated metal ions, ensuring that metal ions flow out less after entering the graphite layer.

After thermal treatment of NZFP/GICs at 900 °C in a muffle furnace, NZF/EG was successfully prepared and it had an excellent expansion volume of 120–160 mL/g. Figure 4 displays the TEM and mapping images of NZF/EG. It is evident that a large number of particles are dispersed on the graphite layers. The lattice fringe of these particles in Figure 4a was observed using high-resolution TEM, as depicted in Figure 4b,c. The (311) lattice plane of ${\mathrm{Ni}}_{0.5}{\mathrm{Zn}}_{0.5}{\mathrm{Fe}}_{2}O_{4}$ corresponds to 0.2529 nm and 0.2526 nm, while the (220) lattice plane of ${\mathrm{Ni}}_{0.5}{\mathrm{Zn}}_{0.5}{\mathrm{Fe}}_{2}O_{4}$ corresponds to 0.2968 nm [28], which verifies the formation of nickel–zinc ferrite. Furthermore, the mapping images of Figure 4a are obtained and shown in Figure 4d–f and indicate that nickel, zinc, and iron elements are evenly distributed on the graphite layers.

Figure 5a displays the XRD pattern of NZF/EG. The diffraction peaks at 26.5° and 54.6° correspond to the (002) and (004) crystal planes of graphite, respectively, indicating that graphite expands along the C-axis and remains composed of graphite microcrystals with a layered structure. Furthermore, the XRD pattern features characteristic diffraction peaks of ${\mathrm{Ni}}_{0.5}{\mathrm{Zn}}_{0.5}{\mathrm{Fe}}_{2}O_{4}$ in the crystal planes of (111), (220), (311), (400), (422), (511), and (440) that match well with the standard JCPDS Card No. 52-0278 [29], suggesting a high crystallinity of NZF. Using Sherrer’s Formula (2) and Equation (3) [30], the crystallite size of NZF at the peak position of (311) was calculated to be 24.2 nm and the lattice parameter was 8.42 Å, which is similar to the previously reported 8.36 Å nano NZF [29]. These results demonstrate the feasibility of the as-proposed synthesis method.

In Figure 5b, the infrared absorbance of NZF/EG is lower compared to that of NZFP/GICs in Figure 4b. This can be attributed to the decomposition of oxygen-containing groups at high temperatures, which is one of the reasons for the expansion of graphite. Apart from the peaks similar to those in NZFP/GICs, the characteristic peak of NZF/EG at 573.9 cm^−1^ corresponds to the stretching vibration mode related to the metal–oxygen absorption band in the crystalline lattice of NZF. This vibration mode usually occurs at the tetrahedral site of the metal lattice and can be observed in the range of 620–550 cm^−1^ [31].
(2)$$D=\frac{K\lambda }{\beta \cos\theta }.$$
where *D* represents the crystallite size, *K* is the Scherer’s constant (with a value of 0.94), *λ* is the X-ray wavelength (1.54056 Å), *β* is the full width at half maximum (FWHM) of the diffraction peak corrected by the instrumental breadth, and *θ* refers to the Bragg angle.
(3)$$a=d_{\mathrm{hkl}}\sqrt{h^{2}+k^{2}+l^{2}}$$
where *a* is the lattice parameter of the particle; $d_{\mathrm{hkl}}$ refers to the interlayer spacing; and *h*, *k*, and *l* represent the Miller indices of (311).

In order to get the distribution of NZF in EG, SEM images of the surface and interior of composites were shown in Figure 6. In Figure 6a, it can be observed that the NZF/EG exhibits a unique worm-like structure and that the surface of the NZF/EG is loose and porous with varying pore sizes. This porous structure not only serves as a loading position for NZF but also enhances the reflection and scattering of electromagnetic waves in graphite [32]. The sample for observing the interior was prepared by cutting the composites along the cross-section in liquid nitrogen. The truncation position and observation position of the sample are shown in Figure 6d,e. From Figure 6c,f, it can be observed that many particles are distributed on the surface and interior of EG. The size of NZF particles inside graphite is larger than that on the surface which is because the stable environment inside graphite makes it easier for NZF to grow and abundant ion sources on the surface of graphite make it easier for NZF to nucleate. EDS spectra show that the main components of these particles are iron, nickel, and zinc, indicating that iron, nickel, and zinc elements enter the interlayers of graphite and generate ferrites between the layers. Thus, NZF is not only distributed uniformly on the surface of EG, but it also exists evenly between the layers of EG.

### 2.2. Distribution of Magnetic Particles in EG Interlayers

#### 2.2.1. Model Building

In order to provide a detailed explanation of the position of magnetic particles in EG layers, the distribution of magnetic particles was simulated by molecular dynamics and certain assumptions were made based on the intercalation structure of NZF/EG. Firstly, it was assumed that the EG layer is flat and that the two graphite sheets are parallel. Secondly, it was assumed that the magnetic particles are spherical and their initial positions are random. In the absence of a magnetic field, magnetic particles are primarily affected by van der Waals forces which arise from interactions between particles and between particles and graphite layers. The van der Waals forces between particles are attractive in nature [33], as indicated by Equation (4). When particles come into contact, in order to avoid the coincidence of particles, the surface of particles is assumed to be a hard surface and the repulsive force between particles is introduced [34]. This repulsive force is characterized by certain features: when two particles come into contact (with the distance between their centers being 2*R*, *R* is the radius of the particle), the repulsive force is balanced with the attraction force. When the distance between particles is greater than 2*R*, the force is negligible, and when it is less than 2*R*, the force is significant. The formula for calculating the repulsive force is given by Equation (5).
(4)$$F_{\mathrm{vdw},\;\mathrm{ij}}=\frac{A_{1}}{6}\frac{d_{i}^{6}}{{\left(h_{\mathrm{ij}}^{2}+2d_{i}h_{\mathrm{ij}}\right)}^{2}{\left(h_{\mathrm{ij}}+d_{i}\right)}^{3}}$$
where $F_{\mathrm{vdw},\;\mathrm{ij}}$ is the van der Waals force between particle *i* and particle *j*, $A_{1}$ is the Hamaker constant between particles, $h_{\mathrm{ij}}$ is the distance between the surface of particle *i* and particle *j*, and $d_{i}$ is the diameter of the particle.
(5)$$f_{\mathrm{rep},\;\mathrm{ij}}=K_{1}\exp[K_{2}(\frac{r_{\mathrm{ij}}}{2R}-1)]\vec{e_{r}}$$
where $f_{\mathrm{rep},\;\mathrm{ij}}$ is the van der Waals force between particle *i* and particle *j*, $K_{1}$ is obtained from the equilibrium condition, and $K_{2}$ is estimated using debugging methods. Based on extensive debugging, the value of $K_{2}$ is determined to be −12 [35]. The $r_{\mathrm{ij}}$ represents the distance between the centers of the particles, while *R* represents the radius of the particle.

The van der Waals force between the particle and the graphite layer is dependent on their distance from each other [36,37]. If the distance between them is greater than the particle radius, the van der Waals force can be calculated using Equation (6). On the other hand, if the distance is less than the particle radius, the calculation formula used is Equation (7). In addition, due to the hard surface of the particle, when the particle makes contact with the graphite layer a repulsive force is created between them [38,39]. This repulsive force is depicted in Equation (8). When the distance between the particle center and the graphite layer equals *R*, this repulsive force is balanced with the attraction force.
(6)$$F_{v,\;z>R}=-\frac{A_{2}}{6}\left(\frac{4R^{3}}{z^{2}{\left(z+2R\right)}^{2}}\right)$$
(7)$$F_{v,\;z<R}=-\frac{A_{2}R}{6z^{2}}$$
where $F_{v,z>R}$ and $F_{v,z<R}$ are van der Waals forces between the particle and the graphite layer, $A_{2}$ is the Hamaker constant between the particle and the graphite layer, and *z* is the distance from the particle center to the graphite layer.
(8)$$f_{r}=K_{3}(\exp[K_{4}(\frac{z_{i}}{2R}-0.5)]-\exp[K_{4}(\frac{{H-z}_{i}}{2R})])$$
where $K_{3}$ is obtained from the equilibrium condition and $K_{4}$ is estimated through debugging methods. After extensive debugging, the value of $K_{4}$ is determined to be −30 [35]. $z_{i}$ represents the distance from the particle center to the plane and *H* represents the distance between the two graphite layers.

In addition to van der Waals forces, magnetic particles are subject to gravity, Brownian forces, and dragging forces. Since the particle size is in the nanometer range, and the simulated environment is at normal temperature, gravity and Brownian forces are ignored. The dragging force is generated by the airfield between graphite layers [35], as shown in Equation (9).
(9)$$F_{t}=D\frac{dx_{i}}{\mathrm{dt}}$$
where $F_{t}$ is the dragging force, *D* is the drag coefficient (*D* = 6π*ηR*), and *η* is the fluid viscosity.

After analyzing the forces acting on particles, Newton’s second law is utilized to describe the distribution of magnetic particles between graphite layers [35,40,41]. Equation (10) expresses the motion of any particle in the *x* direction. Solving Equation (10) provides the particle displacement ∆*x_i_*, as shown in Equation (11). Table 1 presents the values of basic parameters used in the calculation, where particle size and simulated space size were obtained from statistical data. Based on the aforementioned assumptions and equations, the cuboid computing space is established. The *y* direction boundary surface is the graphite layer and periodic boundary conditions are set in the *x* and *z* directions. When the particle moves out of bounds, a particle is added in the opposite direction. The model diagram is shown in Figure 7a.
(10)$$m\frac{d^{2}x_{i}}{dt^{2}}+D\frac{dx_{i}}{\mathrm{dt}}=F_{x,i}$$
(11)$${∆x}_{i}=\frac{F_{x,i}}{D_{i}}\tau +\frac{m_{i}}{D_{i}}\left(v_{i,0}-\frac{F_{x,i}}{D_{i}}\right)\left(1-e^{-D_{i}/m_{i}\tau }\right)$$
where *τ* is the time step, $v_{i,0}$ represents the initial velocity of the particle, and its value is set to 0.

#### 2.2.2. Distribution of Magnetic Particles

The distribution of magnetic particles under different calculation steps is shown in Figure 7. In comparison to the initial state, when the calculation step is 2000, the magnetic particles tend to move towards the *y*–*z* plane (graphite layers). When the calculation step is 20,000, most magnetic particles bond with the graphite layers. However, there are still some magnetic particles in the suspension state which is due to the short simulation time. From the molecular dynamics simulation results, it can be concluded that under the synergy of van der Waals forces, repulsive force, and dragging force, magnetic particles exhibit a tendency to approach each other and contact the graphite layers directly, which avoids the formation of large aggregates and increases the contact area between magnetic particles and graphite layers.

### 2.3. Radar Wave Attenuation Property

The schematic diagram in Figure 8 illustrates the radar wave attenuation process and mechanism of NZF/EG aerosols. Firstly, the electromagnetic waves emitted by the radar will be attenuated and lose energy after entering the NZF/EG aerosols. The part through the NZF/EG aerosols will reach the target surface and be reflected by the target to form echo waves. Subsequently, attenuation occurs again when the echo waves come across aerosols on the way back to the radar receiver [42]. Within NZF/EG aerosols, the attenuation is achieved by multiple absorption and scattering of the incident radar waves. Various loss mechanisms occur during the absorption and scattering processes, such as dipolar polarization and conduction loss on the EG layers; eddy current loss and natural resonance in NZF; and interfacial polarization and exchange resonance at the interfaces between NZF and EG which convert the energy of electromagnetic waves into other forms of energy (heat, kinetic energy, etc.) or reradiate at the same wavelength. It is worth noting that the morphology characterization and molecular dynamics simulation results show that NZF has a large interface contact with the graphite layer which is conducive to the enhancement of interfacial polarization and exchange resonance. Therefore, aerosols formed by NZF/EG composites will have better attenuation performances compared with aerosols formed by pure NZF or EG.

In order to better understand the electromagnetic loss mechanism of as-prepared composites, the real and imaginary parts of the complex permittivity and complex permeability of EG, NZF/EG-0.5, and NZF/EG-1.0 are analyzed in detail. $\varepsilon^{\prime }$ represents the degree of polarization of the materials in the electric field, including interfacial polarization and dipole polarization. The $\varepsilon^{\prime }$ values in Figure 9a decrease with increasing frequency; due to the acceleration of the electric field period with increasing frequency, the dipoles in the materials cannot be quickly reoriented [43]. The $\varepsilon^{\prime }$ values of NZF/EG are greater than that of EG which is due to the enhanced interfacial polarization caused by the introduction of NZF. $\varepsilon^{\prime \prime }$ represents the dielectric loss of the materials, including the polarization loss and conductivity loss. The $\varepsilon^{\prime \prime }$ values exhibit a downward trend which can be explained by the decrease of polarization degree with increasing frequency. The $\varepsilon^{\prime \prime }$ values of NZF/EG are greater than that of EG in Figure 9b due to the interfacial polarization introduced by the embedding of NZF into EG layers which enhances the dielectric loss performance of the composites [44,45]. Compared with NZF/EG-0.5, the $\varepsilon^{\prime \prime }$ values of NZF/EG-1.0 are smaller, which can be explained by the fact that the presence of more NZF reduces the conductivity of EG, thereby reducing conductivity loss. It is worth noting that there are obvious peaks in the $\varepsilon^{\prime \prime }$ curves of NZF/EG at 14.3 GHz and 16 GHz due to the increase in dipolar polarization when the frequency of the applied electric field is the same as the natural vibration frequency of the dipoles in the materials [46].

The dielectric loss of a material consists of conductivity loss, dipolar polarization, and interfacial polarization. On the basis of the Debye dipolar relaxation model, the relationship between $\varepsilon^{\prime }$ and $\varepsilon^{\prime \prime }$ is described by the following equation [47]:(12)$${\left(\varepsilon^{\prime }-\varepsilon_{\infty }\right)}^{2}+{\left(\varepsilon^{\prime \prime }\right)}^{2}={\left(\varepsilon_{s}-\varepsilon_{\infty }\right)}^{2}$$
where $\varepsilon_{\infty }$ is the relative dielectric constant in terms of the high-frequency limit and $\varepsilon_{s}$ is the static dielectric constant. From mathematical Equation (12), it can be deduced that the curve of $\varepsilon^{\prime }$ versus $\varepsilon^{\prime \prime }$ would form a single semicircle called the Cole–Cole semicircle. Each semicircle represents the existence of one Debye relaxation process. The Cole–Cole semicircles of EG, NZF/EG-0.5, and NZF/EG-1.0 are presented in Figure 9c–e, and the dashed circles in Figure 9c–e are used to label the Cole–Cole semicircles in the curve. The Cole–Cole curve of pure EG contains several semicircles. With the incorporation of NZF, more semicircles are obviously observed in curves of NZF/EG-0.5 and NZF/EG-1.0, which indicates that the composites have multiple dielectric polarization relaxation processes that are caused by the delay of induced charges. For as-prepared composites, dipolar polarization occurs at the defects on EG layers. Additionally, the unique intercalation structure of NZF/EG introduces large amounts of NZF–EG interfaces, leading to charge accumulation at the interfaces and causing the interfacial polarization that is also called the Maxwell–Wagner–Sillars effect. In addition to polarization loss, the conductivity loss caused by the movement of charge carriers on EG layers also contributes to electromagnetic wave attenuation.

$\mu^{\prime }$ reflects the magnetization degree of the materials in the magnetic field and its value depends on the saturation magnetization value $M_{s}$ and other factors. EG and $M_{s}$ do not have hysteresis loops because they have diamagnetism; the $M_{s}$ values of NZF/EG-0.5 and NZF/EG-1.0 are 4.92 emu/g and 8.44 emu/g, respectively, which results in the fact that the $\mu^{\prime }$ values of NZF/EG are higher than that of EG in Figure 10a. $\mu^{\prime \prime }$ reflects the magnetic loss performance of materials, including eddy current loss, natural resonance, exchange resonance, etc. [48]. It can be seen from Figure 10b that the $\mu^{\prime \prime }$ values of NZF/EG are higher than that of EG, which can be interpreted as the fact that the introduction of NZF enhances the above magnetic loss mechanism. In addition, the three samples show multiple strong peaks in the range of 7–18 GHz due to the enhanced exchange resonance between NZF and EG when the vibration frequency of composites is same as the external magnetic field [49].

In 2–18 GHz, the magnetic loss of materials is mainly attributed to eddy current loss, natural resonance, and exchange resonance. Generally, the eddy current coefficient $C_{0}$ is used to determine whether eddy current loss plays a major role in magnetic loss, which can be expressed as the following equation [50]:(13)$$C_{0}=\mu^{\prime \prime }{\left(\mu^{\prime }\right)}^{-2}f^{1}=2\pi \mu_{0}d^{2}\delta$$
where $\mu_{0}$ and δ are the vacuum permeability and the electric conductivity, respectively. When $C_{0}$ is constant within a frequency band, it indicates that the magnetic loss is caused by the eddy current effect. Figure 10c shows the $C_{0}$ curves of EG, NZF/EG-0.5, and NZF/EG-1.0. Within the range of 2–18 GHz, the $C_{0}$ value of EG is greater than that of NZF/EG due to the stronger conductivity of EG. The $C_{0}$ value of EG with significant changes indicates that eddy current loss is not the main cause of EG magnetic loss. The $C_{0}$ value of NZF/EG remains almost unchanged in 2.0–14.4 GHz, demonstrating that eddy current loss is the main magnetic loss mechanism, which is not conducive to a microwave entering the absorber and should be suppressed in subsequent research. Meanwhile, the change in the $C_{0}$ value of NZF/EG in 14.4–18.0 GHz is due to the enhanced exchange resonance between NZF and EG, which can be explained from Figure 10b. Therefore, it can be inferred that the magnetic loss of NZF/EG is mainly related to eddy current loss and exchange resonance.

The attenuation constant (α) is often used to reflect the dissipative capacity of materials to the incident electromagnetic wave. According to electromagnetic wave transmission theory, the attenuation constant can be expressed as Equation (14) [51,52,53].
(14)$$\alpha =\frac{\sqrt{2}\pi f}{c}\sqrt{(\mu^{\prime \prime }\varepsilon^{\prime \prime }-\mu^{\prime }\varepsilon^{\prime })+[{\left(\mu^{\prime \prime }\varepsilon^{\prime \prime }-\mu^{\prime }\varepsilon^{\prime }\right)}^{2}+{\left(\mu^{\prime \prime }\varepsilon^{\prime }+\mu^{\prime }\varepsilon^{\prime \prime }\right)}^{2}{]}^{0.5}}$$
where $\varepsilon^{\prime }$ and $\varepsilon^{\prime \prime }$ are the real and imaginary parts of the complex permittivity of materials and $\mu^{\prime }$ and $\mu^{\prime \prime }$ are the real and imaginary parts of the complex permeability of materials. *f* represents the external electromagnetic field frequency while c denotes the speed of light.

Figure 11 shows that the α value increases with increasing frequency, indicating that EG and NZF/EG have high attenuation performances in high frequency ranges. The α values of NZF/EG-0.5 are significantly greater than that of EG, indicating that the addition of NZF greatly enhances the radar wave attenuation ability of EG in the range of 2–18 GHz. Meanwhile, the α values of NZF/EG-0.5 are greater than that of NZF/EG-1.0 due to the good retention of the dielectric property of the graphite layer, while the area of the heterogeneous interface is increased at a ratio of 0.5. Therefore, the ratio of ferrite to graphite is an important factor affecting the attenuation performance of EG. In the following research work, we will explore the appropriate ratio of ferrite to graphite to obtain the relationship between ratio and the attenuation performance.

## 3. Materials and Methods

### 3.1. Materials

Natural flake graphite (50 mesh, purity > 99.9%, referred to as “NG”) was purchased from the Shandong Qingdao Nanshu graphite mine (Qingdao, China). As for the 65.0~68.0 wt% nitric acid, 99.5 wt% acetic acid, 25.0 wt% ammonia, and $\mathrm{Ni}(NO_{3}{)}_{2}\cdot 6H_{2}O$ (98.0 wt%), they were purchased from Beijing Tongguang Fine Chemical Co., Ltd. (Beijing, China). Potassium permanganate (99.5 wt%) was purchased from the Tianjin Damao Chemical Reagent Factory (Tianjing, China). $\mathrm{Zn}(NO_{3}{)}_{2}\cdot 6H_{2}O$ (99.0 wt%) was supplied by Beijing Chemical Works (Beijing, China). $\mathrm{Fe}(NO_{3}{)}_{3}\cdot 9H_{2}O$ (98.5 wt%) was supplied by Jiangsu Aikang Biomedical Research and Development Co., Ltd. (Nanjing, China). All chemicals were of analytic grade and were used directly as obtained. All experiments were carried out using deionized water.

### 3.2. Synthesis of NZFP/GICs

NZFP/GICs were obtained by the preparation of graphite oxide (GO) and graphite intercalation compounds. GO was prepared by chemical oxidation. Firstly, 1 g of NG was dispersed ultrasonically in 6 mL of nitric acid for 10 min. Then, 0.4 g of potassium permanganate and 3 mL of acetic acid were added and the mixed solution was stirred at 30 °C for 1 h. After washing the composites to pH = 6 and performing centrifugation, GO was obtained.

Graphite intercalation compounds were synthesized by chemical coprecipitation. Firstly, a 0.5 mol/L nitrate solution was formulated by dissolving $\mathrm{Ni}(NO_{3}{)}_{2}\cdot 6H_{2}O$, $\mathrm{Zn}(NO_{3}{)}_{2}\cdot 6H_{2}O$, and $\mathrm{Fe}(NO_{3}{)}_{3}\cdot 9H_{2}O$ with water. The intercalation reaction solution was prepared by mixing the nitrate solution with the above graphite oxide and stirring it at 60 °C for 30 min. Then, ammonia water was added to pH = 8–9 and the reaction continued for 30 min. The Ni–Zn ferrite precursor intercalated graphite compounds (NZFP/GICs) were obtained by washing the composites 4–5 times and drying them in the oven at 50 °C for 24 h.

The dosages of the above nitrate were determined by the mass ratios of NZF and NG and the mole ratios of nickel, zinc, and iron were 1:1:4. The composites with mass ratios of NZF to NG $\left(m_{\mathrm{NZF}}/m_{\mathrm{NG}}\right)$ including 0, 0.5, and 1.0 were prepared and separately named EG, NZF/EG-0.5, and NZF/EG-1.0. The mass ratio of the composites, not specified in this paper, was 1.0.

### 3.3. Synthesis of NZF/EG

Ni–Zn ferrite intercalated EG (NZF/EG) was prepared via the thermal treatment of NZFP/GICs at 900 °C. Firstly, the muffle furnace (SX-5-12, 7 L, 220 V, 5 kW, Teste, Tianjin, China) was heated to 900 °C and held at this temperature for 1 h. Next, the quartz beaker (150 mL) was preheated for 5 min. Then, 0.5 g of NZFP/GICs was rapidly added to the preheated quartz beaker, the beaker was removed after 5–10 s, and NZF/EG could be obtained.

### 3.4. Characterization

The morphologies of NZFP/GICs and NZF/EG were investigated using the scanning electron microscope (SEM, ZEISS Gemini 300, Carl Zeiss, Oberkochen, Germany) and transmission electron microscopy (TEM, JEM-F200, JEOL, Tokyo, Japan) operating at an acceleration voltage of 200 kV. High-resolution TEM was applied to measure the lattice spacings of the composites. The compositions of the samples were characterized using energy dispersive X-ray spectroscopy (EDS, OXFORD XPLORE30, OXIG, Oxford, UK). The Fourier transform infrared (FTIR, Frontier, Perkin Elmer, Waltham, MA, USA) spectra of the composites mixed with KBr powder were recorded from 500 cm^−1^ to 4000 cm^−1^ at a resolution of 4 cm^−1^. To identify the crystalline structure and phase composition of the composites, the X-ray diffractometer (XRD, SmartLab SE, Rigaku, Tokyo, Japan) with Cu-Kα radiation (λ = 0.154056 nm) was used to obtain the diffraction peak of composites in the range of 10° to 80°. The electromagnetic parameters $\left(\varepsilon_{r}=\varepsilon^{\prime }-{j\varepsilon }^{\prime \prime }\;,\;\mu_{r}=\mu^{\prime }-{j\mu }^{\prime \prime }\right)$ of the composites were acquired using the Vector Network Analyzer (VNA, E5071C, Agilent, Frankfurt, Germany) over a frequency range of 2–18 GHz by mixing composites and paraffin wax with a mass ratio of 1:9. The magnetic property of the composites was measured at room temperature on a vibrating sample magnetometer (VSM, 7404, Lake Shore, OH, USA) under the maximum magnetic field of 2 T. 

## 4. Conclusions

In summary, the study successfully prepares NZF intercalated EG through oxidation, intercalation, coprecipitation, and subsequent thermal treatment at 900 °C. The morphology and phase characterization results show that the precursor of NZF is successfully intercalated into the graphite interlayers. The as-prepared composite forms conductive magnetic composites after thermal treatment at 900 °C and the ferrite has a large contact area with the graphite sheet. The α values of NZF/EG-0.5 are higher, indicating it has a better radar wave attenuation ability compared to EG and NZF/EG-1.0 in the range of 2–18 GHz. This paper presents a new and innovative method for the in situ formation of magnetic particles intercalated with EG for attenuating the centimeter-wave radar, which introduces a new idea for the development of passive interference materials.

## Figures and Tables

**Figure 1 molecules-28-04128-f001:**
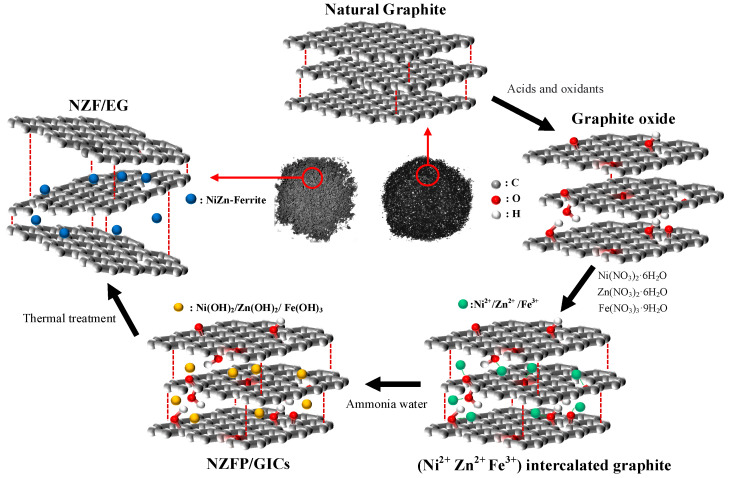
Schematic illustration of the preparation of NZF/EG.

**Figure 2 molecules-28-04128-f002:**
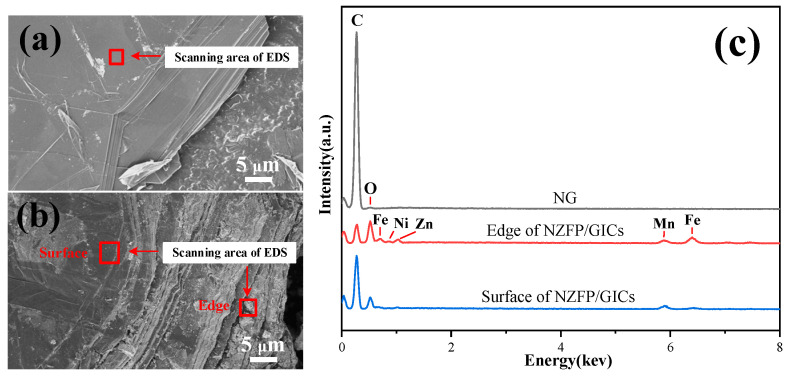
SEM images of (**a**) NG and (**b**) NZFP/GICs and (**c**) EDS spectra of NG and NZFP/GICs.

**Figure 3 molecules-28-04128-f003:**
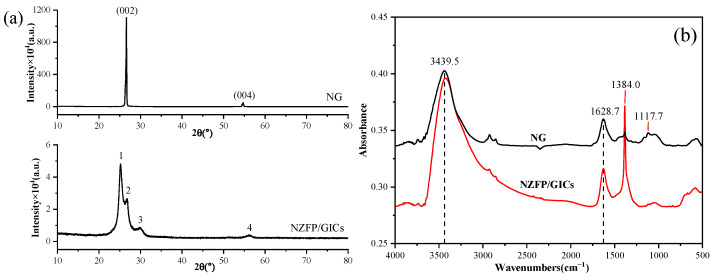
(**a**) XRD patterns and (**b**) FTIR spectra of NG and NZFP/GICs.

**Figure 4 molecules-28-04128-f004:**
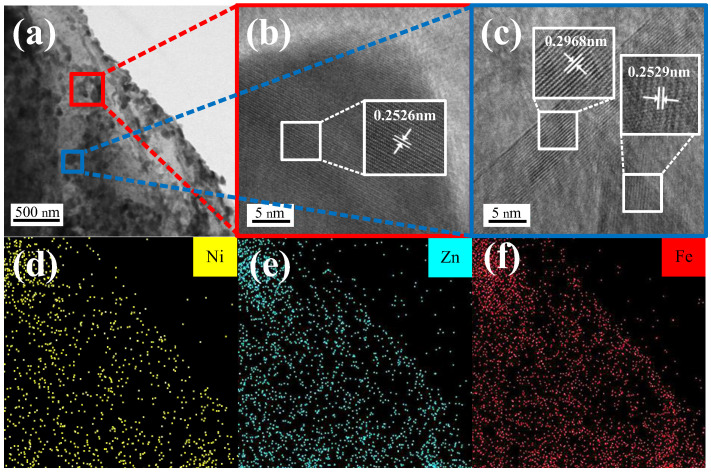
(**a**) TEM images of NZF/EG, (**b**,**c**) HR-TEM images of NZF/EG, (**d**–**f**) TEM mapping images of NZF/EG.

**Figure 5 molecules-28-04128-f005:**
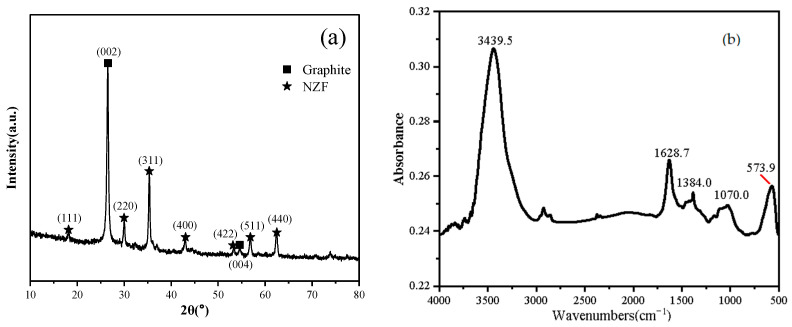
(**a**) XRD pattern and (**b**) FTIR spectrum of NZF/EG.

**Figure 6 molecules-28-04128-f006:**
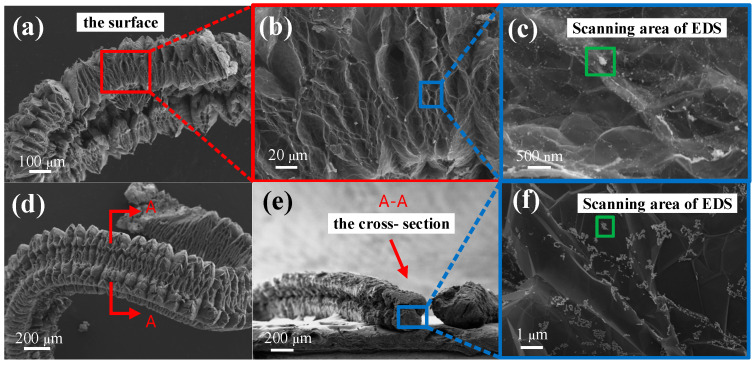
SEM images of the (**a**–**c**) surface and (**d**–**f**) cross-section of NZF/EG and EDS spectra of the (**g**) surface and (**h**) cross-section of NZF/EG.

**Figure 7 molecules-28-04128-f007:**
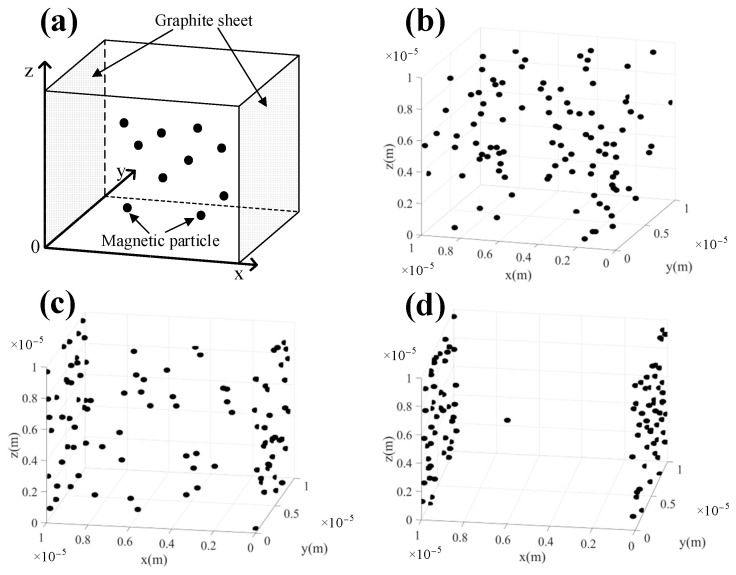
(**a**) Model diagram of magnetic particles in graphite layers, (**b**) the initial positions of the particles, and (**c**,**d**) the positions of the particles after program running 2000 and 20,000 steps.

**Figure 8 molecules-28-04128-f008:**
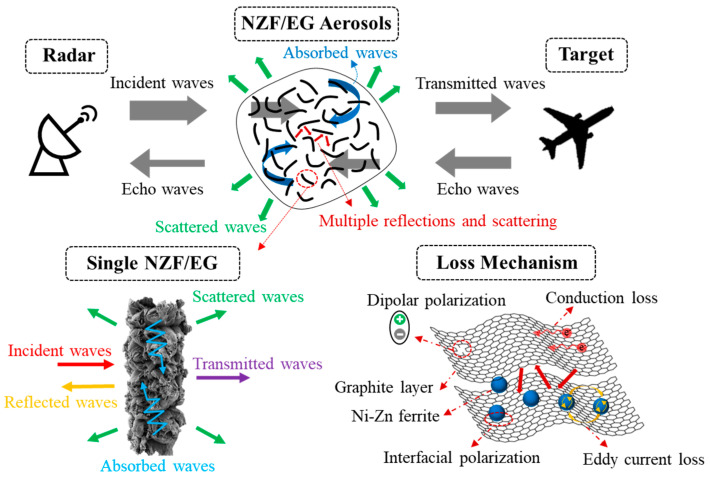
Schematic diagram of the radar wave attenuation process and mechanism of NZF/EG aerosols.

**Figure 9 molecules-28-04128-f009:**
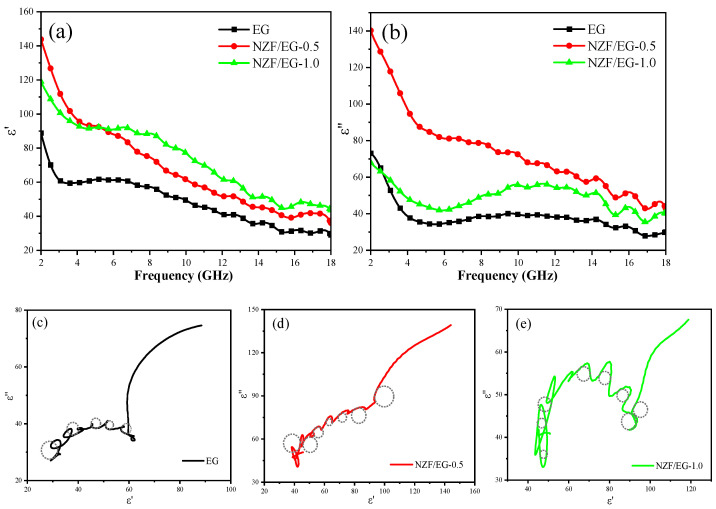
(**a**) Real part and (**b**) imaginary part of the complex permittivity of EG, NZF/EG-0.5, and NZF/EG-1.0. Cole–Cole semicircles of (**c**) EG, (**d**) NZF/EG-0.5, and (**e**) NZF/EG-1.0.

**Figure 10 molecules-28-04128-f010:**
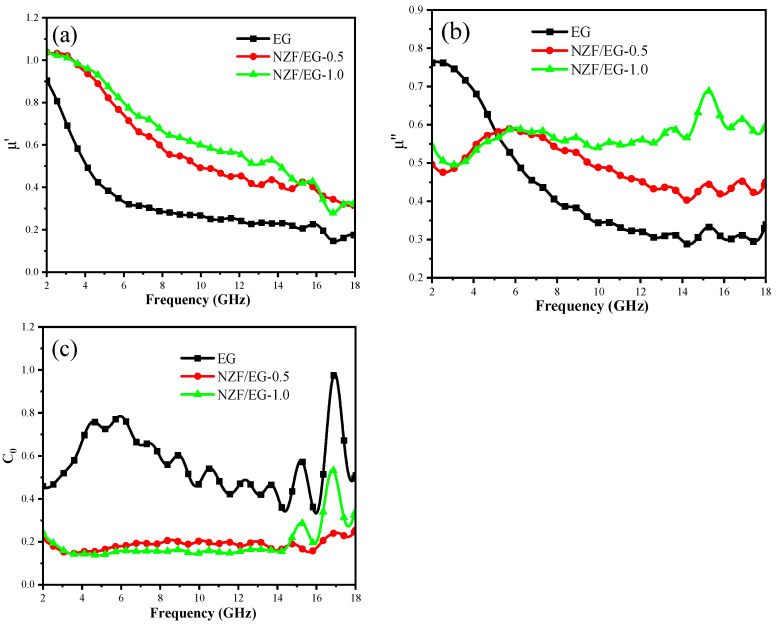
(**a**) The real part and (**b**) the imaginary part of the complex permeability and (**c**) $C_{0}$ values of EG, NZF/EG-0.5, and NZF/EG-1.0.

**Figure 11 molecules-28-04128-f011:**
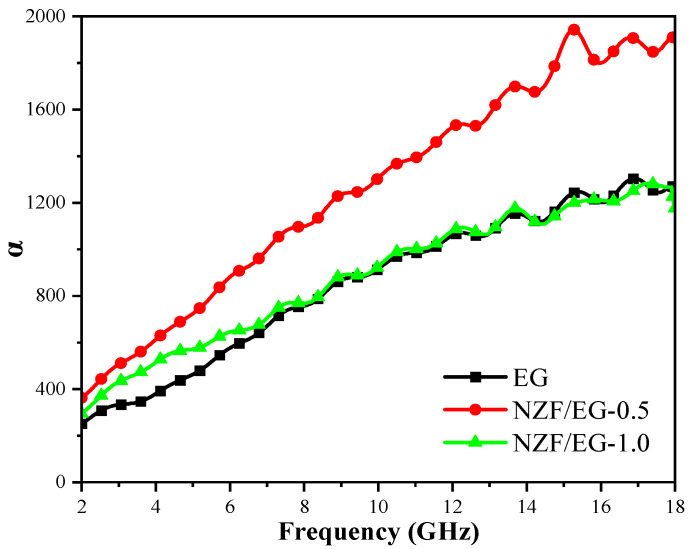
Attenuation constant of EG, NZF/EG-0.5, and NZF/EG-1.0.

**Table 1 molecules-28-04128-t001:** Basic parameters.

Basic Parameter	Symbol	Value
Particle size	*a*	300 nm
Particle number	*N*	100
The length of the simulated space	length	10 μm
The width of the simulated space	width	10 μm
The height of the simulated space	height	10 μm
Hamaker constant between particles	$A_{1}$	2 × 10^−19^
Hamaker constant between particles and graphite layers	$A_{2}$	2 × 10^−19^
Air viscosity	η	$1.837\times 10^{-4}\;Pa\cdot s$
Time step	τ	$2\times 10^{-8}\;s$

## Data Availability

Not applicable.

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
