# Peer review of "The In Situ Preparation of Ni–Zn Ferrite Intercalated Expanded Graphite via Thermal Treatment for Improved Radar Attenuation Property"

_molecules, 2023, doi:10.3390/molecules28104128_

Round 1

Reviewer 1 Report

The manuscript proposes a new preparation method of Ni-Zn ferrite intercalated in composites of expanded graphite with possible applications in the field of radar wave attenuation.

The paper presents the preparation method, the characterization of the resulting compounds by XRD, SEM, FT-IR, molecular dynamic simulation of magnetic particles.

In my opinion, the work can be accepted with the following minor corrections:

-In formula (2) it should be specified that β is width at half maximum corrected by the instrumental breadth.

- Line 150, grain size should be replaced with crystallite size. Crystallite size is determined from XRD and grain size is obtained from SEM.

- Crystallite size should be compared with grain size (crystallite size should be smaller or equal to grain size).

- The number of graphene layers could be evaluated with the formula n=D/d where D is the crystallite sizes determined from the Scherrer relation and d the distance between the crystallographic planes determined from the Bragg relation.

Author Response

Authors would like to express our most sincere gratitude to you for your effort and patience in reviewing our manuscript.  According to your advice, we amended the relevant parts in the manuscript. Please see the attachment.

Reviewer 2 Report

In the present work, authors reported the synthesis of Ni-Zn ferrite intercalated EG by using thermal treatment of Ni-Zn ferrite precursor intercalated graphite, and then investigated their structural, morphological, and attenuation property. Results indicated the composite with the NZF ratio at 0.5 showed the best radar wave attenuation ability due to that the dielectric property of the graphite layers was well retained while the area of the heterogeneous interface is increased. This work has certain reference function as an applied research. However, some issues should be addressed.

1, The increased distance might be proved by the XRD results at the lower angle range of less than 10° or others? In XRD, which angle did θ refer to? Please clarify it.

2, It was unable to provide a satisfactory modelling of the attenuation mechanism (s): To what is due microwave absorption (magnetism, polarization, relaxation, resonance ...)? This fundamental issue is not all answered.

3, It was said that “Due to the decrease of polarization degree, the ε'' values decrease with increasing frequency. It can be seen from Figure. 9(b) that the ε'' values of NZF/EG are greater than that of EG, indicating that the polarization loss plays a more important role”. Since the contributions of each polarization loss and conductive loss were not evaluated, we can not draw a conclusion that polarization loss plays a more important role. Otherwise, the real values of each loss might be given?

4, Some key and important research results in attenuation field should be mentioned and cited so that we can provide a solid background and progress to the readers, such as Journal of Materials Chemistry C, 2016, 4, 9738; Small, 2018, 14, 1800987; ACS Applied Materials & Interfaces, 2017, 9, 16404; Nano-Micro Letters, 2022, 14, 173.

5, The authors fabricated ferrite particles on EGs by using a moderate chemical hydrothermal reaction. The control of content of loading ferrite is very important. I am wondering how its loading is controlled and how the content of loading ferrite. I suggest you make a component analysis, such as spectroscopic analysis.

6, Herein, the preparation method of absorbers for attenuation is hot pressing with wax. As far as I am concerned, it is just to meet the test conditions of coaxial line, and how to achieve the strength requirements in practical application. In addition, I noticed that the volume ratio of filler in wax was 10%. So the dielectric and magnetic properties were very amazing. Please give more details.

Author Response

(The authors gave the same response as above.)
